# CoupAlign: Coupling Word-Pixel with Sentence-Mask Alignments for Referring Image Segmentation

**Zicheng Zhang**[1]* **Yi Zhu**[2]* **Jianzhuang Liu**[2] **Xiaodan Liang**[3] **Ke Wei**[1]†

[1]Xi'an Jiaotong University
[2]Noah's Ark Lab, Huawei Technologies
[3]Sun Yat-sen University

zzc1999@stu.xjtu.edu.cn   {zhuyi36,liu.jianzhuang}@huawei.com
xdliang328@gmail.com   wei.ke@xjtu.edu.cn

## Abstract

Referring image segmentation aims at localizing all pixels of the visual objects described by a natural language sentence. Previous works learn to straightforwardly align the sentence embedding and pixel-level embedding for highlighting the referred objects, but ignore the semantic consistency of pixels within the same object, leading to incomplete masks and localization errors in predictions. To tackle this problem, we propose **CoupAlign**, a simple yet effective multi-level visual-semantic alignment method, to couple **sentence-mask alignment** with **word-pixel alignment** to enforce object mask constraint for achieving more accurate localization and segmentation. Specifically, the Word-Pixel Alignment (WPA) module performs early fusion of linguistic and pixel-level features in intermediate layers of the vision and language encoders. Based on the word-pixel aligned embedding, a set of mask proposals are generated to hypothesize possible objects. Then in the Sentence-Mask Alignment (SMA) module, the masks are weighted by the sentence embedding to localize the referred object, and finally projected back to aggregate the pixels for the target. To further enhance the learning of the two alignment modules, an auxiliary loss is designed to contrast the foreground and background pixels. By hierarchically aligning pixels and masks with linguistic features, our CoupAlign captures the pixel coherence at both visual and semantic levels, thus generating more accurate predictions. Extensive experiments on popular datasets (e.g., RefCOCO and G-Ref) show that our method achieves consistent improvements over state-of-the-art methods, e.g., about 2% oIoU increase on the validation and testing set of RefCOCO. Especially, CoupAlign has remarkable ability in distinguishing the target from multiple objects of the same class. Code will be available at https://gitee.com/mindspore/models/tree/master/research/cv/CoupAlign.

## 1 Introduction

Taking an image and a natural language sentence as input, a referring image segmentation (RIS) model is required to predict a mask for the object described by the sentence. Beyond semantic segmentation which assigns each pixel in an image with a label from a fixed word set, referring image segmentation needs to localize the object pixels indicated by the language expression, which is of arbitrary length and involves an open-world vocabulary, such as object names, attributes, positions,

---

*Equal contribution.
†Corresponding author.

36th Conference on Neural Information Processing Systems (NeurIPS 2022).

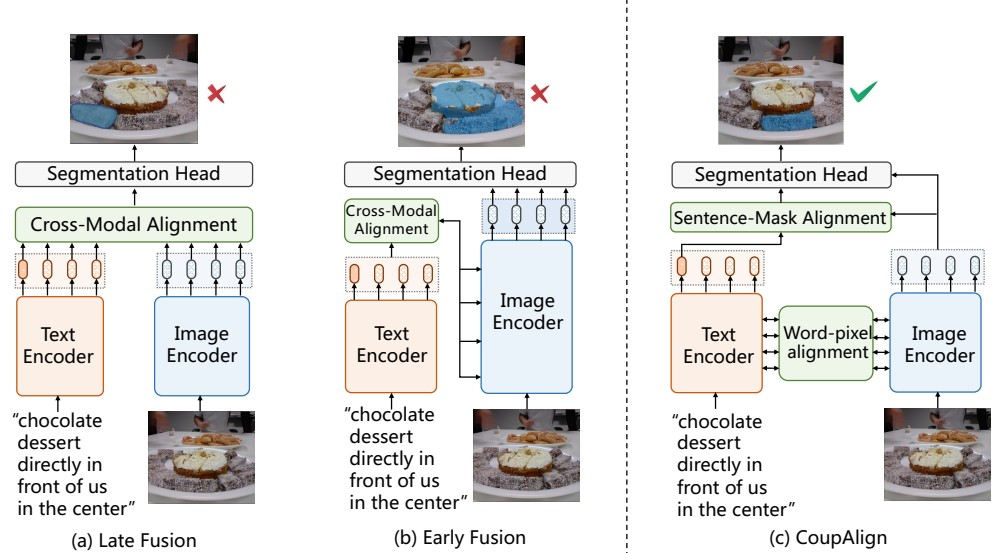

Figure 1: Illustration of the cross-modal alignment (Late/Early Fusion) in existing RIS methods and our CoupAlign that enables hierarchical alignment at word-pixel and sentence-mask levels.

etc. Therefore, this task is the foundation of many applications in the real world, such as image editing [26, 34], autonomous driving [4, 38], and language-based human-robot interaction [35, 2].

The main challenge of RIS is how to align the given language expression with visual pixels for highlighting the target. Early works [16, 7] typically follow the paradigm that first extracts visual and language features from the corresponding encoder and then fuses them for language-pixel alignment. Based on the aligned feature, a decoder is employed to predict a mask. Recently, Transformer-based vision-language models [20, 24, 22] have shown great potential in aligning visual and language features from fine to coarse levels, e.g., pixel, concept, region and image from visual view, and e.g., word, phrase, sentence, paragraph from language perspective. Inspired by this, extensive works [17, 7, 41, 11, 44] have been investigated to take the advantage of the remarkable cross-modal alignment ability of Transformer architectures for aligning language with pixels more effectively. Some works [7, 41] seek to improve the cross-modal alignment of the encoded vision and language features, as shown in Fig. 1(a). Their cross-modal alignment mainly occurs at the sentence-pixel level. Furthermore, other works [11, 44] fuse the language information into the pixel embeddings at early stages of the image encoder, as shown in Fig. 1(b). The alignment happens more earlier than before, starting from the word-pixel level which is more fine-grained.

Though effective, in Fig. 1(a) and (b), we can also observe many undesirable pixel fragments and localization errors in their failure cases. The reason lies behind these issues may be that the pixel-level alignment (e.g., word-pixel and sentence-pixel level) connects each pixel embedding to language independently, and ignores the visual and semantic consistency among pixels in an object. Therefore, the predicted masks may miss some target pixels or spill over to other objects. To deal with this problem, we propose to regard the pixels of an object as a whole and explicitly introduce object mask constraints for the cross-modal alignment, at both the fine-grained and the coarse-grained levels.

Specifically, in this paper, we propose **CoupAlign** that couples sentence-mask alignment with word-pixel alignment to enforce the object mask constraint for improving the localization and segmentation performance of RIS. The **Word-Pixel Alignment (WPA)** module enables cross-modal interaction at the intermediate layers of the vision and language encoders. Then a set of mask proposals are calculated based on the word-pixel aligned features to indicate the possible objects in an image. The **Sentence-Mask Alignment (SMA)** module learns to weight the masks using the sentence embedding to localize the referred object, and the weighted masks are projected back to provide mask signals for effectively aggregating the pixels of the target. To further enhance the word-pixel and sentence-mask alignments, we design an auxiliary loss that forces the pixels of the target to get close and the pixels not belonging to the target to get away. Our CoupAlign captures the pixel coherence at both visual

and semantic levels, and hierarchically aligns the pixels and masks with language information. By doing this, CoupAlign generates more accurate mask predictions for RIS.

To validate the effectiveness of our method, we conduct extensive experiments on popular datasets (RefCOCO [18] and G-ref [32]). It is shown that CoupAlign outperforms state-of-the-art RIS methods by significant margins, e.g, about 2% oIoU increase on the validation and testing sets of RefCOCO. Besides, CoupAlign shows remarkable ability in distinguishing the target from multiple (even greater than twenty) objects of the same class, for example, localizing one person in a team photo using a language query like "middle row second kid from right".

The main contributions of this paper are summarized as follows:

- We tackle the challenging problem of RIS via integrating the mask-aware constraint into the pixel-level semantic alignment for capturing the pixel coherence within objects.

- We propose CoupAlign, a simple yet effective alignment framework, to couple sentence-mask alignment with word-pixel alignment to hierarchically align pixels and masks with linguistic information.

- We conduct comprehensive experiments to demonstrate that our CoupAlign predicts more accurate object locations and masks and significantly outperforms state-of-the-art RIS methods on popular benchmarks.

## 2 Related Work

**Referring Image Segmentation** aims to segment the objects referred by natural language expressions. RIS Methods can be summarized into two categories. One focuses on cross-model alignment and representation learning [16, 25, 11]. Another views this problem as a language-conditioned visual reasoning [43, 14], which utilizes a dependency parsing tree to iteratively select the proposal regions.

Due to the breakthrough of Transformer [8] in the computer vision field and its remarkable fusion power for multi-modality, many works [17, 7, 44] begin to explore the cross-model alignment based on Transformer. MDETR [17] shows that simply concatenating vision and language features and then feeding them into the Transformer encoder and decoder can achieve surprising results in many vision-language downstream tasks. VLT [7] is the first work to introduce the Transformer architecture to referring segmentation. It designs a query generation module to utilize the contextual information of images to enrich language expressions and improve robustness. EFN [11] and LAVT [44] insert language-aware attention modules into the image encoding process to achieve early fusion of cross-modal features. MaiL [25] abandons the dual-stream encoders and adopts an unified cross-model Transformer to achieve word-pixel alignment. CRIS [41] focuses on how to transfer CLIP's [36] strong image-text alignment ability to benefit sentence-pixel alignment required by RIS. CRIS uses a Transformer decoder with word queries and a sentence-pixel contrastive loss to achieve this goal.

The above mentioned RIS methods mainly focus on aligning each pixel to word or sentence independently while ignoring the pixel coherence of the target object, thus resulting in inaccurate masks and localization. In contrast, our method couples sentence-mask alignment with word-pixel alignment to hierarchically align pixels and masks with linguistic information, and improving the segmentation and localization accuracy.

**Vision Language Modeling** targets at seeking an unified representation for the two modalities. From the aspect of architecture, vision language models can be divided into single-stream and dual-stream architecture structures. Single-stream models such as VisualBERT [22], Uniter [3], Uniencoder-vl [21] and Oscar [24], utilizes an unified encoder based on self-attention to fuse the vision and language embeddings. For dual-stream models such as CLIP [36], ALIGN [15], FILIP [45] and GLIP [23], they adopt two unimodality encoders to encode vision and language embeddings and utilizes the dot product to align the outputs. Others like ViLBERT [29] and LXMERT [37] uses two self-attention-based encoders to model intra-modality interactions and employ cross-attention to model cross-modality interactions. Inspired by GLIP [23], we adopt a hybrid encoder design, where dual encoders are used to extract vision and language features and the word-pixel alignment is realized at the intermediate stages of the encoders.

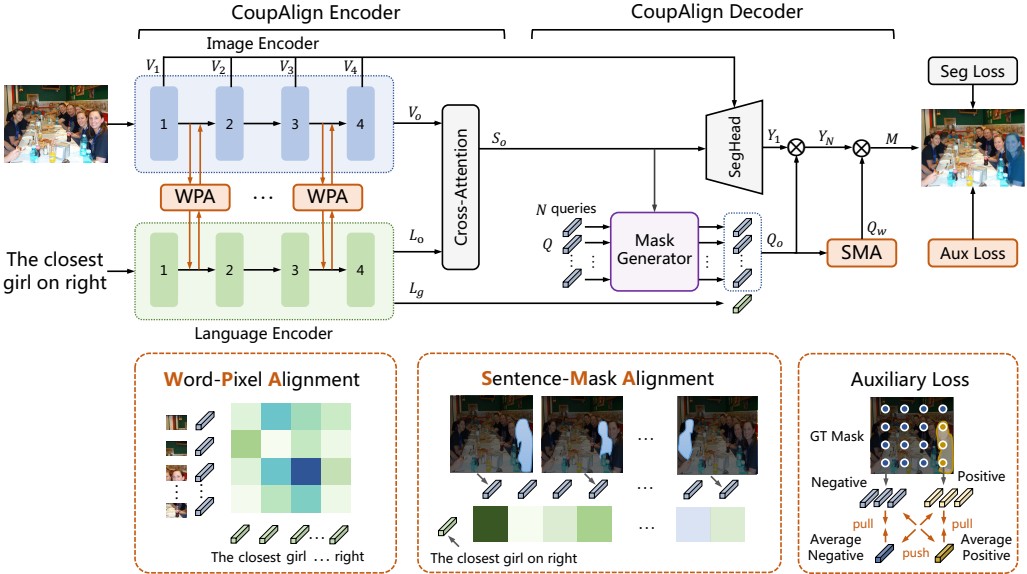

Figure 2: Architecture of our CoupAlign. Given an image and a language sentence, the image features $V_o$ can be extracted from the image encoder, and the word-level features $L_o$ and sentence feature $L_g$ are extracted from the language encoder. The Word-Pixel Alignment (WPA) module enables cross-modal interactions at each encoder stage. Based on the word-pixel aligned features $S_o$, the mask generator produces $N$ mask embeddings $Q_o$. The Sentence-Mask Alignment (SMA) weights $Q_o$ using $L_g$ and projects the mask signals back to $Y_N$, and finally generates the prediction $M$.

## 3  Methodology

### 3.1  Overview

The overall architecture of our CoupAlign is shown in Fig. 2. It is designed following the encoder-decoder paradigm. The CoupAlign encoder consists of an image encoder $f_v$ and a language encoder $f_l$ for extracting vision and language embeddings, a Word-Pixel Alignment (WPA) module that enables the cross-modal interaction at the intermediate layers of the visual and language encoders, and a cross attention module for cross-modal fusion after encoders. The CoupAlign decoder includes a mask generator module that produces $N$ segment proposals, a segmentation head (SegHead) that upsamples the pixel-level embeddings, and a Sentence-Mask Alignment module (SMA) to weight the masks via the sentence and then project back to the pixel-level embeddings to enforce object mask constraint. Together with the segmentation loss, an auxiliary loss that compares foreground pixels with background ones to enhance the hierarchical aligning at word-pixel and sentence-mask levels.

### 3.2  CoupAlign Encoder

**Image Encoder**. Given an image $I \in \mathbb{R}^{H \times W \times 3}$, we adopt Swin Transformer [27] as the image encoder backbone to extract multi-level visual features. We utilize the features from the 1st–4th stages of Swin Transformer, denoted as $\{V_i \in \mathbb{R}^{H_i \times W_i \times C_i}\}_{i=1}^4$, where $C_i$, $H_i$, $W_i$ are the number of channels, the height, and the width of the feature map at the $i$-th stage, respectively.

**Language Encoder.** We use BERT [6] with 12 layers as our language encoder backbone. Given a referring expression $\{w_i\}_{i=1}^T$, where $T$ is the length of the expression, we first use the BERT tokenizer based on WordPiece embeddings [42] to get the word embeddings $E \in \mathbb{R}^{T \times D}$, where $D$ is the hidden feature dimensionality of BERT. Then the word embeddings $E$ are fed into the hidden layers of BERT to extract language features. In order to make word-pixel alignment with vision features from Swin Transformer, we aggregate every 3 hidden layers into a stage, so that we can have the same stage number of language features as the vision features, which are denoted as $\{L_i \in \mathbb{R}^{T \times D}\}_{i=1}^4$. We also utilize the output [CLS] token as the global semantic representation of the referring expression, which is denoted as $L_g \in \mathbb{R}^{1 \times D}$.

**Word-Pixel Alignment (WPA).** For segmentation task, we need to obtain a fine-grained segmentation mask, which means that low-level features like words in sentences and edges and shapes of objects in images are crucial for finding correct segments. The cross-modal alignment at early encoder stage is proved to be effective for referring image segmentation [7, 44]. Their early alignments are in single-direction of language to vision. In contrast, our WPA module is implemented with bidirectional cross attention to achieve both language-to-vision and vision-to-language alignment, as:

$$V_i', L_i' = \text{BiAttn}(V_i, L_i), \quad i \in \{1, \dots, 4\}, \tag{1}$$

$$V_i' \leftarrow \text{Gate}(V_i'), \quad L_i' \leftarrow \text{Gate}(L_i'), \tag{2}$$

$$V_{i+1} = \text{SwinStage}_i(V_i + L_i'), \quad L_{i+1} = \text{BERTStage}_i(L_i + V_i'), \tag{3}$$

where $\text{SwinStage}_i(\cdot)$ represents the Stage $i$ defined in Swin Transformer. $\text{BERTStage}_i(\cdot)$ represents the Stage $i$ of BERT. The cross-modality interaction is achieved by the bidirectional cross attention (BiAttn), which computes the context vectors of one modality by attending to the other modality:

$$\hat{V}_i = V_i W_i^v, \quad \hat{L}_i = L_i W_i^l, \quad Attn_i = \hat{V}_i \hat{L}_i^T / \sqrt{d}, \tag{4}$$

$$L_i' = \text{softmax}(Attn_i) L_i \hat{W}_i^l, \quad V_i' = \text{softmax}(Attn_i^T) V_i \hat{W}_i^v, \tag{5}$$

where $d$ is the dimension of the joint embedding space, and $W_i^v \in \mathbb{R}^{C_i \times d}$, $W_i^l \in \mathbb{R}^{D \times d}$, $\hat{W}_i^l \in \mathbb{R}^{d \times C_i}$, $\hat{W}_i^v \in \mathbb{R}^{d \times D}$ are all projection matrices.

In order to prevent the fused features $V_i'$ and $L_i'$ from overwhelming the original signals $V_i$ and $L_i$ respectively. We design a Gate network [44] to control the fused information flow:

$$\text{Gate}(F_i) = \text{MLP}(F_i) \odot F_i, \tag{6}$$

where $F_i$ is the fused feature from BiAttn, $\odot$ indicates element-wise multiplication, and MLP is a two-layer perceptron with the first layer being a linear layer followed by the ReLU activation [33] and the second layer being a linear layer followed by a hyperbolic tangent activation.

**Cross Attention.** To better capture high-level semantics and generate the pixel-level fused features, we use a plain multi-head attention layer [40] to fuse the high-level vision and language features from the image and language encoders. As illustrated in Fig. 2, first, we obtain the high-level vision features $V_o \in \mathbb{R}^{H_o \times W_o \times C_o}$ from the output of the image encoder and the high-level language features $L_o \in \mathbb{R}^{T \times D}$ from the language encoder. Then we map the vision and language features to $\mathbb{R}^D$. After that we concatenate them to produce a $F_o \in \mathbb{R}^{(H_o W_o + T) \times D}$, which is fed into the cross attention layer. We add learnable positional emdeddings $e_p \in \mathbb{R}^{H_o \times W_o \times D}$ to the vision features before the concatenation. Finally the output of the cross attention layer is added with the original input vision features to produce a low-resolution feature map $S_o \in \mathbb{R}^{H_o \times W_o \times C_o}$, as:

$$V_o' = V_o W_o^v + e_p, \quad L_o' = L_o W_o^l, \tag{7}$$

$$S_o = \text{CrossAttn}(F_o) + V_o, \quad F_o = [V_o'; L_o'], \tag{8}$$

where $W_o^v \in \mathbb{R}^{C_o \times D}$ and $W_o^l \in \mathbb{R}^{D \times D}$ are projection matrices and [ ; ] denotes concatenation.

### 3.3 CoupAlign Decoder

**Mask Generator.** To provide object mask constraints for the word-pixel alignment, we first design a mask generator that produces $N$ segment proposals based on the output feature of the CoupAlign encoder. The generator is built upon a 6-layer transformer decoder. It takes $S_o \in \mathbb{R}^{H_o \times W_o \times C_o}$ and $N$ learnable queries $Q \in \mathbb{R}^{N \times d_q}$ as input, where $d_q$ is the hidden dimensionality of the transformer decoder, and outputs the hidden feature $Q_o \in \mathbb{R}^{N \times d_q}$ of the $N$ segment tokens, as:

$$Q_o = \text{MaskGenerator}(Q, S_o). \tag{9}$$

**Segmetation Head (SegHead).** To upsample the pixel-level features for generating final segmentation maps, we construct a segmentation head that takes the CoupAlign encoder output feature $S_o$ and the multi-level visual features $\{V_i\}_{i=1}^4$ as input, and obtains output features as:

$$\begin{cases} Y_5 = S_o, \\ Y_i = \text{Up}(\rho(Y_{i+1})) + \gamma(V_i), \quad i = 4, 3, 2, 1, \end{cases} \tag{10}$$

where $\rho$ is a two-layer convolution network where each layer performs a $3\times3$ convolution followed by ReLU and batch normalization, Up denotes $2\times$ bilinear upsampling, $\gamma$ is a $1\times1$ convolution that maps the vision features to $\mathbb{R}^{d_s}$. We take $Y_1 \in \mathbb{R}^{\frac{H}{4} \times \frac{W}{4} \times d_s}$ as the output of SegHead.

**Sentence-Mask Alignment (SMA).** After generating mask proposals and obtaining upsampled pixel-level embeddings, the sentence-mask alignment assigns each mask a score according to the language query and then project the weighted masks back to the pixel-level feature maps to introduce object mask constraint for word-pixel alignment. At first, We project $Q_o$ and $Y_1$ to $\hat{Q}_o \in \mathbb{R}^{N \times D}$ and $\hat{Y}_1 \in \mathbb{R}^{\frac{H}{4} \times \frac{W}{4} \times D}$ via learnable weights $W^Q \in \mathbb{R}^{d_q \times D}$ and $W^Y \in \mathbb{R}^{d_s \times D}$. We use the mask embeddings $\hat{Q}_o$ and the sentence representation $L_g$ generated from the language encoder to compute attention weight $Q_w \in \mathbb{R}^{1 \times N}$. Higher weight in $Q_w$ indicate that the corresponding mask is most likely to contain the object referred by the language. Then, $\hat{Q}_o$ and $\hat{Y}_1$ are multiplied to get $N$ mask predictions $Y_N \in \mathbb{R}^{N \times \frac{H}{4} \times \frac{W}{4}}$. Finally, $Y_N$ is multiplied by $Q_w$ to generate the final mask prediction $M \in \mathbb{R}^{\frac{H}{4} \times \frac{W}{4}}$, which is further upsampled to the input image size. This procedure is represented as:

$$Q_w = \text{softmax}(sim(L_g, \hat{Q}_o)), \tag{11}$$

$$M = Y_N \otimes Q_w, \quad Y_N = \hat{Y}_1 \otimes \hat{Q}_o, \tag{12}$$

where $sim(\cdot)$ is cosine similarity and $\otimes$ denotes the broadcast tensor multiplication.

## 3.4 Loss Functions

**Segmentation Loss.** In the context of referring image segmentation, each pixel in the image $I$ is discriminated into background or foreground. Thus this task can be viewed as a pixel-wise binary classification problem. Specifically, we use bilinear interpolation to upsample $M$ to the original image size, obtaining $M'$. Let $m'_i$ and $\hat{m}_i$ be the values of pixel $i$ from $M'$ and $\hat{M}$ (ground truth mask) respectively. Then the segmentation loss are implemented via a binary cross entropy loss as:

$$L_j^{\text{Seg}} = -\frac{1}{H \times W} \sum_{i=1}^{H \times W} \hat{m'}_i \log(\sigma(m_i)) + (1 - \hat{m}_i)\log(1 - \sigma(m'_i)), \tag{13}$$

where $\sigma$ denotes sigmoid function, and $j$ is the image index in a training batch.

**Auxiliary Loss.** The word-pixel and sentence-mask alignment collaborate in a hierarchically learning manner to enforce object mask constraint for improving referring image segmentation. We adopt a pixel-level contrastive loss as auxiliary to further enhance the learning of the two alignments and enable the aligning consistency at both the fine-grained and coarse-grained level. This loss forces the pixels within the ground truth masks to get closer, and the rest pixels to get away. Therefore the foreground pixels are aggregated together to generate more accurate predictions. Specifically, we resize the ground truth mask $\hat{M}$ to the same shape as the segmentation map $Y_1$. Let $y_i$ be the vector of $Y_1$ in pixel $i$. Then we use the resized mask to divide positive and negative samples in each image. For pixel $i$, if $\hat{m}_i$ is 0, then $y_i$ belongs to the negative set $\mathcal{N}$ and is denoted as $y_i^-$; otherwise $y_i$ belongs to the positive set $\mathcal{P}$ and is denoted as $y_i^+$. Then we average the negative vectors and the positive vectors to get the negative prototype $\hat{y}^-$ and the positive prototype $\hat{y}^+$. Afterward we adopt InfoNCE [39] to compute the contrastive loss, as:

$$L_{P2N}^{\text{Aux}} = -\frac{1}{|\mathcal{P}|} \sum_{y_i^+ \in \mathcal{P}} \frac{\exp(y_i^+ \cdot \hat{y}^+/\tau)}{\exp(y_i^+ \cdot \hat{y}^+/\tau) + \sum_{y_k^- \in \mathcal{N}} \exp(y_i^+ \cdot y_k^-/\tau)}, \tag{14}$$

$$L_{N2P}^{\text{Aux}} = -\frac{1}{|\mathcal{N}|} \sum_{y_i^- \in \mathcal{N}} \frac{\exp(y_i^- \cdot \hat{y}^-/\tau)}{\exp(y_i^- \cdot \hat{y}^-/\tau) + \sum_{y_k^+ \in \mathcal{P}} \exp(y_i^- \cdot y_k^+/\tau)}, \tag{15}$$

$$L_j^{\text{Aux}} = L_{P2N}^{\text{Aux}} + L_{N2P}^{\text{Aux}}, \tag{16}$$

where $\tau$ is the temperature parameter. Finally, the total loss for the training is that:

$$L = \frac{1}{B} \sum_{j=1}^{B} (L_j^{\text{Seg}} + \lambda L_j^{\text{Aux}}), \tag{17}$$

where $B$ is the batch size, $\lambda$ is a hyper-parameter.

| | Backbone | RefCOCO | | | RefCOCO+ | | | G-Ref | | ReferIt |
|---|---|---|---|---|---|---|---|---|---|---|
| | | val | test A | test B | val | testA | testB | val (U) | test (U) | test |
| MCN [31] | Darknet-53 | 62.44 | 64.20 | 59.71 | 50.62 | 54.99 | 44.69 | 49.22 | 49.40 | - |
| BRINet [12] | ResNet-101 | 60.98 | 62.99 | 59.21 | 48.17 | 52.32 | 42.11 | - | - | 63.46 |
| CMPC [13] | ResNet-101 | 61.36 | 64.53 | 59.64 | 49.56 | 53.44 | 43.23 | - | - | 65.53 |
| LSCM [14] | ResNet-101 | 61.47 | 64.99 | 59.55 | 49.34 | 53.12 | 43.50 | - | - | 66.57 |
| CGAN [30] | ResNet-101 | 64.86 | 68.04 | 62.07 | 51.03 | 55.51 | 44.06 | 51.01 | 51.69 | - |
| BUSNet [43] | ResNet-101 | 63.27 | 66.41 | 61.39 | 51.76 | 56.87 | 44.13 | - | - | - |
| EFN [11] | ResNet-101 | 62.76 | 65.69 | 59.67 | 51.50 | 55.24 | 43.01 | - | - | 66.70 |
| LTS [16] | DarkNet-53 | 65.43 | 67.76 | 63.08 | 54.21 | 58.32 | 48.02 | 54.40 | 54.25 | - |
| VLT [7] | DarkNet-53 | 65.65 | 68.29 | 62.73 | 55.50 | 59.20 | 49.36 | 52.99 | 56.65 | - |
| ReSTR [19] | ViT-B-16 | 67.22 | 69.30 | 64.45 | 55.78 | 60.44 | 48.27 | 54.48 | - | 70.18 |
| CRIS [41] | ResNet-101 | 70.47 | 73.18 | 66.10 | 62.27 | 68.08 | 53.68 | 59.87 | 60.36 | - |
| LAVT [44] | Swin-B | 72.73 | 75.82 | 68.79 | 62.14 | **68.38** | 55.10 | 61.24 | 62.09 | - |
| CoupAlign (ours) | Swin-B | **74.70** | **77.76** | **70.58** | 62.92 | 68.34 | **56.69** | 62.84 | 62.22 | 73.28 |

Table 1: Comparison with SOTA methods on the oIoU metric on RIS datasets.

# 4 Experiments

## 4.1 Settings

**Dataset.** We evaluate our method on four widely-used datasets: RefCOCO [46], RefCOCO+[46], G-Ref [32] and ReferIt [18]. RefCOCO consists of 142,209 referring expressions for 50,000 objects in 19,994 images and RefCOCO+ has 141,564 referring expressions for 49,856 objects in 19,992 images. G-Ref was collected on Amazon Mechanical Turk in a non-interactive setting which consists of 85,474 referring expressions for 54,822 objects in 26,711 images. ReferIt has 130,525 referring expressions in 19,894 images which are collected from IAPR TC-12 [9]. RefCOCO+ is disallowed from using location words in referring expressions. The languages used in RefCOCO and RefCOCO+ tend to be more concise and less flowery than the languages used in the G-Ref. The expressions in G-Ref tend to be longer with an average of 8.43 words, which makes it more challenging compared with RefCOCO and RefCOCO+. Expressions in ReferIt are usually shorter than the other datasets.

**Metrics.** We adopt three common used metrics in referring image segmentation: the overall intersection-over-union (oIoU), the mean intersection-over-union (mIoU) and prec@X. The oIoU measures as the ratio between the total intersection area and the total union area of all test samples. By default, we use this metric to compare CoupAlign with other works. The mIoU is the average of the IoU over all test samples. prec@X measures the percentage of test images with an IoU score higher than a threshold $\epsilon$, where $\epsilon \in \{0.5, 0.7, 0.9\}$.

**Implementation Details.** We implement our model using the MindSpore Lite tool [1]. We choose BERT$_{BASE}$ [6] with 12 layers and hidden size 768 as our language encoder and use the official pre-trained weights. The image encoder layers are initialized with the classification weights pre-trained on ImageNet-22K [5]. The number of queries of the mask generator $N$ is 100. The rest of the weights in our model are randomly initialized. We adopt AdamW [28] optimizer with weight decay 0.01. We adopt the polynomial learning rate decay schedule and set the initial learning rate to 3e-5, the end learning rate to 1.5e-5 and the max decay epoch to 25. Our model is trained for 50 epochs with batch size 16. The images are resized to $448 \times 448$ without specific data augmentation and the maximum sentence length is set to 30. The weight of the auxiliary loss $\lambda$ is set to 0.1.

## 4.2 Comparison with State-of-the-Arts

**Results on RefCOCO.** In Tab. 1, we compare CoupAlign with state-of-the-art (SOTA) RIS methods by oIoU metric. The language expressions provided by RefCOCO contain many words of positional relationships, e.g., "The closet girl on the right", which requires the RIS models to not only capture the pixel coherence within objects, but also the spatial relationship among multiple objects of the same class. Compared with the SOTA method LAVT [44], CoupAlign achieves better performance with absolute oIoU increases of 1.97%, 1.94%, and 1.79% on the validation, testA, and testB sets of RefCOCO, respectively.

**Results on RefCOCO+.** As shown in the Tab. 1, the performance of CoupAlign on RefCOCO+ is comparable to or better than LAVT which is the most recent state-of-the-art method (published in CVPR2022). The improvements on RefCOCO+ are less significant than those on G-Ref and

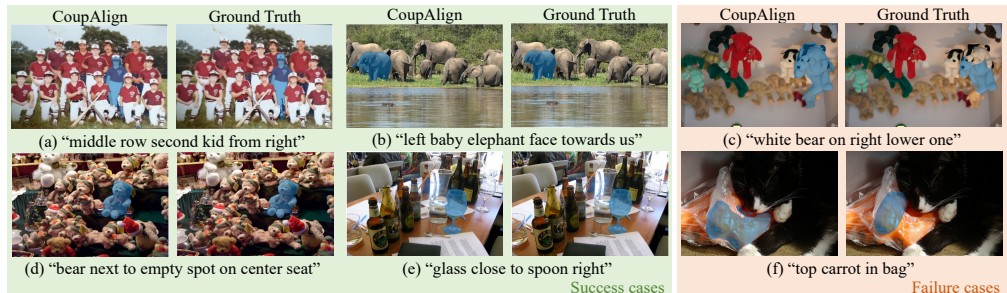

Figure 3: Visualization examples of the success cases and failure cases of CoupAlign.

| IoU Range / Method | [0,0.1) | [0.1,0.2) | [0.2,0.3) | [0.3,0.4) | [0.4,0.5) |
|---|---|---|---|---|---|
| LAVT [44] | 330 | 43 | 64 | 82 | 113 |
| CoupAlign (ours) | 271 | 47 | 44 | 62 | 94 |

Table 2: Analysis of localization ability.

| Num of mask | mIoU | prec@0.5 | prec@0.7 | prec@0.9 |
|---|---|---|---|---|
| 20 | 74.77 | 85.46 | 76.22 | 30.91 |
| 50 | 74.98 | 85.67 | 76.20 | 31.88 |
| 100 | 75.49 | 86.40 | 77.59 | 32.40 |

Table 3: Ablation of different mask numbers.

RefCOCO. This is because the language captions in RefCOCO+ hardly contain the descriptions of the relative or absolute spatial locations (e.g., "closest" or "right bottom") of the target object in images, and the effectiveness of our method mainly lies in the ability of localizing objects in such challenging scenarios, e.g., "middle row second kid from right" in Fig. 3.

**Results on G-Ref.** The language annotations of G-Ref have more complex language structure with more words, e.g., "chocolate dessert directly in front of us in the center", which requires more fine-detailed word-pixel alignment, more discriminative sentence-mask alignment, and a more comprehensive understanding of the sentence. As shown in Tab. 1 on G-Ref (UMD partition), our CoupAlign outperforms LAVT by 1.6% and 0.13% on the validation and test sets, respectively. The consistent improvements on all the evaluated datasets demonstrate the capability of our CoupAlign in modeling hierarchical cross-modal alignment at pixel and mask levels.

**Analysis of Localization Ability.** If the IoU between a mask prediction and the Ground Truth mask is smaller than 0.5, we consider the prediction as a failure case for localization. In Tab. 2, we can see that our CoupAlign has fewer failure cases than LAVT in most IoU ranges. Especially, in the IoU range of [0,0.1), the number of failure cases of CoupAlign decreases the most, which further validates the remarkable ability of CoupAlign.

**Mask Number $N$.** From Tab. 3, we find that the mask number $N$ in mask generator is correlated to the model performance. To balance the computational cost and performance, $N$ is set to 100.

## 4.3 Visualization

**Success cases and failure cases.** In Fig. 3, we show some representative visualization examples of success predictions and failure cases of CoupAlign. The examples are from the RefCOCO val. set. We can see that CoupAlign works well in the scenes where crowded objects have similar color and context and stay close to each other. However, if the boundary of the object is blurred, or there exists occlusion, the segmentation prediction is roughly accurate, but not precise enough.

**Word-Pixel and Sentence-Mask Alignment.** To validate whether our WPA and SMA modules can perform accurate and consistent alignments, we visualize the intermediate attention maps of the alignment modules in Fig. 4. It can be observed that our WPA module accurately highlights the most relevant pixels corresponding to the word. Note that the vocabulary of RIS language annotations is very large compared to the small fixed label set of semantic segmentation. Surprisingly, our CoupAlign can not only capture the semantic meaning of words in various types, e.g., nouns, positions, and attributes, but also distinguish synonyms, e.g., child, man, and lady, which are roughly defined as "person" in many semantic segmentation datasets like PASCAL VOC [10]. As for the SMA module, we show three masks with their attention weights ranked in descending order. The mask with the highest weight is the most accurate one and the masks with lower weights are less accurate. Fig. 4(right) demonstrates that our SMA module provides the effective mask constraint for benefiting the hierarchical aligning at pixel and mask levels.

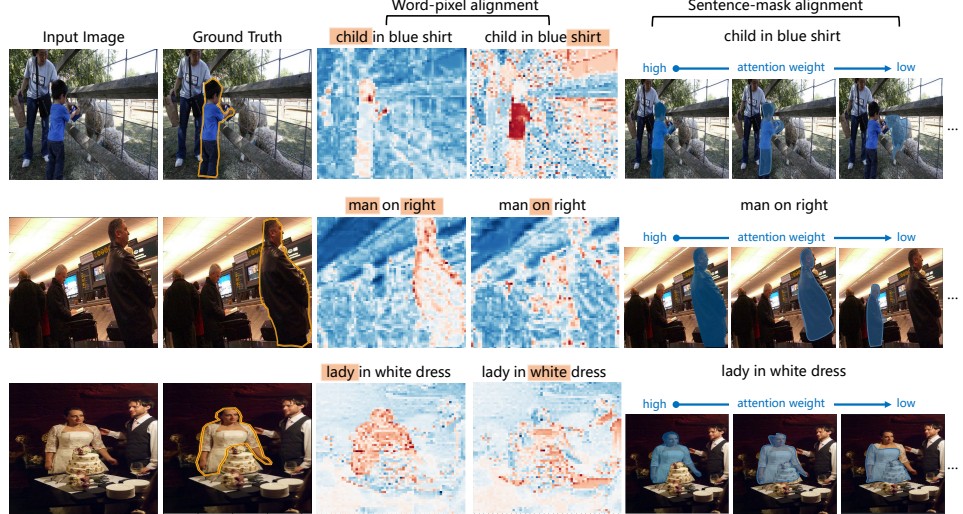

Figure 4: Visualization examples of our word-pixel alignment and sentence-mask alignment.

| Bi-WPA | Uni-WPA | SMA | Aux Loss | oIoU | mIoU | prec@0.5 | prec@0.7 | prec@0.9 |
|--------|---------|-----|----------|------|------|----------|----------|----------|
| | | ✓ | ✓ | 70.43 | 69.61 | 80.03 | 68.91 | 27.47 |
| | ✓ | ✓ | ✓ | 72.70 | 73.44 | 84.07 | 72.60 | 29.02 |
| ✓ | | | ✓ | 73.02 | 73.85 | 84.86 | 74.62 | 29.25 |
| ✓ | | ✓ | | 73.70 | 74.21 | 85.32 | 75.31 | 30.14 |
| ✓ | | ✓ | ✓ | **74.70** | **75.49** | **86.40** | **77.59** | **32.40** |

Table 4: Ablation studies of our WPA module, SMA module, and auxiliary loss.

| WPA's position | stage 1-4 | stage 1,2 | stage 3,4 | stage 4 | stage 3 | stage 2 | stage 1 |
|----------------|-----------|-----------|-----------|---------|---------|---------|---------|
| oIoU | **74.70** | 72.74 | 73.93 | 73.61 | 72.53 | 72.63 | 72.59 |
| mIoU | **75.49** | 73.87 | 74.88 | 74.68 | 73.47 | 73.48 | 73.97 |

Table 5: Ablation study of WPA modules' number and position.

## 4.4 Ablation Study

**Effect of WPA Types.** Our WPA module enables cross-modal interaction at both low-level and high-level encoding stages. In Tab. 4, Bi-WPA indicates our WPA which is implemented via bidirectional cross attention, Uni-WPA represent unidirectional attention only from language to image features, as in LAVT [44]. The performance dramatically drops by 4.3% (74.70% vs. 70.43%) on oIoU when WPA is removed. And also when we replace our Bi-WPA with Uni-WPA, the oIoU result drops by 2% (74.70% vs. 72.70%). These results demonstrate the effectiveness of our WPA module.

**Effect of WPA's Number and Position.** In our experiment, we used four WPA modules, two of which are in the early encoding stage and the other two are in the late encoding stage. To study the effects of the numbers of WPA modules, we first conduct two baseline models that alternatively remove two WPA at early or late encoding stages. As shown in the Tab. 5, when we remove the last two WPA modules the performance drops about 2%, and when we remove the first two WPA modules the performance drops about 0.8%. These results validate the effectiveness of WPA modules at both early and late stages and indicate that the latter WPA modules play a more important role in our modules. Then, we only use one WPA module, which is inserted at different encoding stages. The WPA module at the 4-th stage is more effective than those inserted at other stages.

**Effect of SMA.** In Tab. 4, when SMA is removed, the oIoU result decreases by 1.7% (74.70% vs. 73.02%), since the cross-modal alignment becomes less accurate without mask constraints.

**Effect of Auxiliary Loss.** From the last two columns of Tab. 4, we observe a performance drop of 1% (74.70% vs. 73.70%) on oIoU when the auxiliary loss is disabled, which verifies the ability of the loss in enhancing the word-pixel and sentence-mask alignments.

# 5 Conclusion

To tackle the practical yet challenging problem of referring image segmentation (RIS), we propose CoupAlign which is a simple yet effective cross-modal alignment framework that couples sentence-mask alignment with word-pixel alignment to enforce mask-aware constraint for referring segmentation. The word-pixel alignment module enables fine-grained cross-modal interactions at the early encoding stage. The sentence-mask alignment module uses the sentence features to weight the mask proposals generated based on the word-pixel aligned features, and then projects the mask constraints back to guide the word-pixel alignment. The auxiliary contrastive loss that compares among pixel pairs to further enhance the hierarchical alignments at word-pixel and sentence-mask levels. Benefiting from the collaboration of the alignment learning at different levels, CoupAlign captures both visual and semantic coherence of pixels within the referred object, and significantly outperforms state-of-the-art RIS methods. Especially, CoupAlign has surprising ability in localizing the target from crowded objects of the same class, showing great potential in segmenting natural language referred objects in real-world scenarios.

# 6 Acknowledgment

This work was supported in part by National Key Research and Development Project of China under Grant No. 2020AAA0105600, National Natural Science Foundation of China under Grant No. 62006182, and the Fundamental Research Funds for the Central Universities under Grant No. xzy012020017, Guangdong Province Basic and Applied Basic Research (Regional Joint Fund-Key) Grant No.2019B1515120039, Guangdong Outstanding Youth Fund (Grant No. 2021B1515020061). We gratefully acknowledge the support of MindSpore, CANN (Compute Architecture for Neural Networks) and Ascend AI Processor used for this research.

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
