# CoupAlign: Coupling Word-Pixel with Sentence-Mask Alignments for Referring Image Segmentation

**Zicheng Zhang**[1]* **Yi Zhu**[2]* **Jianzhuang Liu**[2] **Xiaodan Liang**[3] **Ke Wei**[1]†

[1]Xi'an Jiaotong University
[2]Noah's Ark Lab, Huawei Technologies
[3]Sun Yat-sen University

zzc1999@stu.xjtu.edu.cn    {zhuyi36,liu.jianzhuang}@huawei.com
xdliang328@gmail.com    wei.ke@xjtu.edu.cn

## 1 Ablation Study

**Effect of Backbones.** To demonstrate the effectiveness of our approach, we change the image backbone of CoupAlign to different networks, like Resnet101 [3] and Darknet53 [9], and evaluate it on the RefCOCO validation set. In Tab. 1, we compare our results with the methods using Resnet101 as the image backbone. In Tab. 2, we compare the methods using Darknet53. The results show that CoupAlign still suppresses previous methods when using the same image backbone, which indicates that our CoupAlign is compatible with popular backbones.

| Method | oIoU | prec@0.5 | prec@0.6 | prec@0.7 | prec@0.8 | prec@0.9 |
|---|---|---|---|---|---|---|
| LSCM [5] | 61.47 | 73.95 | 69.58 | 62.59 | 49.61 | 20.63 |
| CMPC [4] | 61.36 | 71.27 | 64.44 | 55.03 | 39.28 | 12.89 |
| EFN [2] | 62.76 | 73.95 | 69.58 | 62.59 | 49.61 | 20.63 |
| LAVT [11] | 68.10 | 80.55 | 75.25 | 67.27 | 55.05 | 25.37 |
| CoupAlign (Ours) | **68.93** | **81.26** | **76.54** | **69.69** | **57.07** | **27.44** |

Table 1: Comparison with different methods using Resnet101.

| Method | oIoU | prec@0.5 | prec@0.6 | prec@0.7 | prec@0.8 | prec@0.9 |
|---|---|---|---|---|---|---|
| LTS [6] | 65.43 | 75.16 | 69.51 | 60.74 | 45.17 | 14.41 |
| VLT [1] | 65.65 | 76.20 | - | - | - | - |
| CoupAlign (ours) | **68.16** | **81.02** | **76.35** | **69.53** | **56.91** | **25.95** |

Table 2: Comparison with different methods using Darknet53.

**Ablation of WPA module.**

In our experiment, we use four WPA modules, two of which are in the early encoding stage and the other two are in the late encoding stage. To study the effects of the numbers of WPA modules, we first conduct two baseline models that alternatively remove two WPA at early or late encoding stages. As shown in the following table, when we remove the last two WPA modules the performance drops about 2% (74.70% vs. 72.74% oIoU, 75.49% vs. 73.87% mIoU), and when we remove the first two WPA modules the performance drops about 0.8% (74.70% vs. 73.61% oIoU, 75.49% vs.

---

*Equal contribution.
†Corresponding author.

74.68% mIoU). These results validate the effectiveness of WPA modules at both early and late stages and indicate that the latter WPA modules play a more important role in our modules. Then, we only use one WPA module, which is inserted at different encoding stages. As shown in Tab. 3, the WPA module at the 4-th stage is more effective than those inserted at other stages.

| WPA's number | WPA's position | oIoU | mIoU |
|---|---|---|---|
| 4 | stage 1,2,3,4 | 74.70 | 75.49 |
| 2 | stage 1,2 | 72.74 | 73.87 |
| 2 | stage 3,4 | 73.93 | 74.88 |
| 4 | stage 4 | 73.61 | 74.68 |
| 3 | stage 3 | 72.63 | 73.48 |
| 2 | stage 2 | 72.53 | 73.47 |
| 1 | stage 1 | 72.59 | 73.97 |

Table 3: Ablation study of WPA modules.

## 2 Evaluation on more datasets

**ReferIt.** ReferIt [7] dataset is generated in interactive style like RefCOCO [12], but collects image from IAPR TC-12 [7]. As shown in Tab. 4, the performance of CoupAlign on ReferIt test set is higher than the state-of-the-art method ReSTR, which is published in CVPR2022.

| Method | test |
|---|---|
| LSCM [5] | 66.57 |
| EFN [2] | 66.70 |
| ReSTR [8] | 70.18 |
| CoupAlign | **73.28** |

Table 4: Comparison of oIoU results on the ReferIt dataset.

**RefCOCO+.** As shown in Tab. 5, the performance of CoupAlign on RefCOCO+ is comparable to or better than LAVT which is the most recent state-of-the-art method (published in CVPR2022). The improvements on RefCOCO+ are less significant than those on G-Ref and RefCOCO. This is because the language captions in RefCOCO+ hardly contain the descriptions of the relative or absolute spatial locations (e.g., "closest" or "right bottom") of the target object in images, and the effectiveness of our method mainly lies in the ability of localizing objects in such challenging scenarios, such as "right first bottom fridge" and "middle row second kid from right" in Fig.3 of our paper.

| Method | val | testA | testB |
|---|---|---|---|
| ReSTR [8] | 55.78 | 60.44 | 48.27 |
| CRIS [10] | 62.27 | 68.08 | 53.68 |
| LAVT [11] | 62.14 | **68.38** | 55.10 |
| CoupAlign(Ours) | **62.92** | 68.34 | **56.69** |

Table 5: Comparison of oIoU results on the RefCOCO+ dataset.

## 3 Reproducibility

Our code is reproducible and can be implemented based on several open-source repositories[3][4][5][6][7][8] following the method description as well as implementation details in Section 3 and Section 4 of our paper.

## 4 Code Release

According to the code authorization rule of our institution, we are not allowed to directly attach the code in the submission. We will apply for a code release license after our paper is accepted.

## 5 Potential Negative Societal Impacts

Our method has no ethical risk on datasets usage and privacy violation because all the datasets and tools are publicly available and transparent.

## 6 Limitations

Although CoupAlign achieves remarkable performance on referring image segmentation, it still does not segment finely enough in some scenes. As shown in Fig. 1, when the boundary of the object is blurred, or there exists occlusion, the segmentation prediction is roughly accurate, but not precise enough. We believe that by adding the boundary enhancement module in the future, our method will perform better in these scenarios.

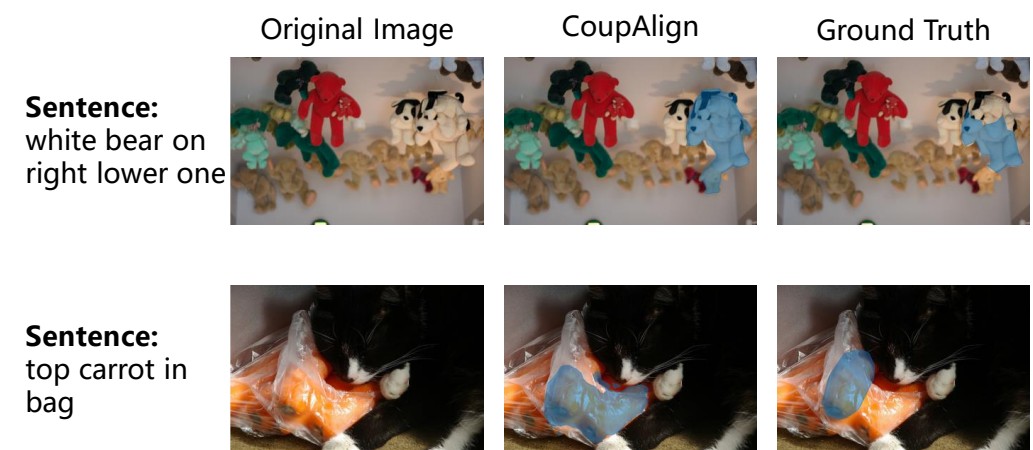

Figure 1: Visualization of failure cases of CoupAlign.

## 7 More Visualization Results

In Fig. 2, we present more visual comparisons with LAVT.

---

[3]https://github.com/mindspore-ai/mindspore

[4]https://github.com/open-mmlab/mmdetection

[5]https://github.com/microsoft/Swin-Transformer

[6]https://github.com/huggingface/transformers

[7]https://github.com/pytorch/vision

[8]https://github.com/zichengsaber/LAVT-pytorch

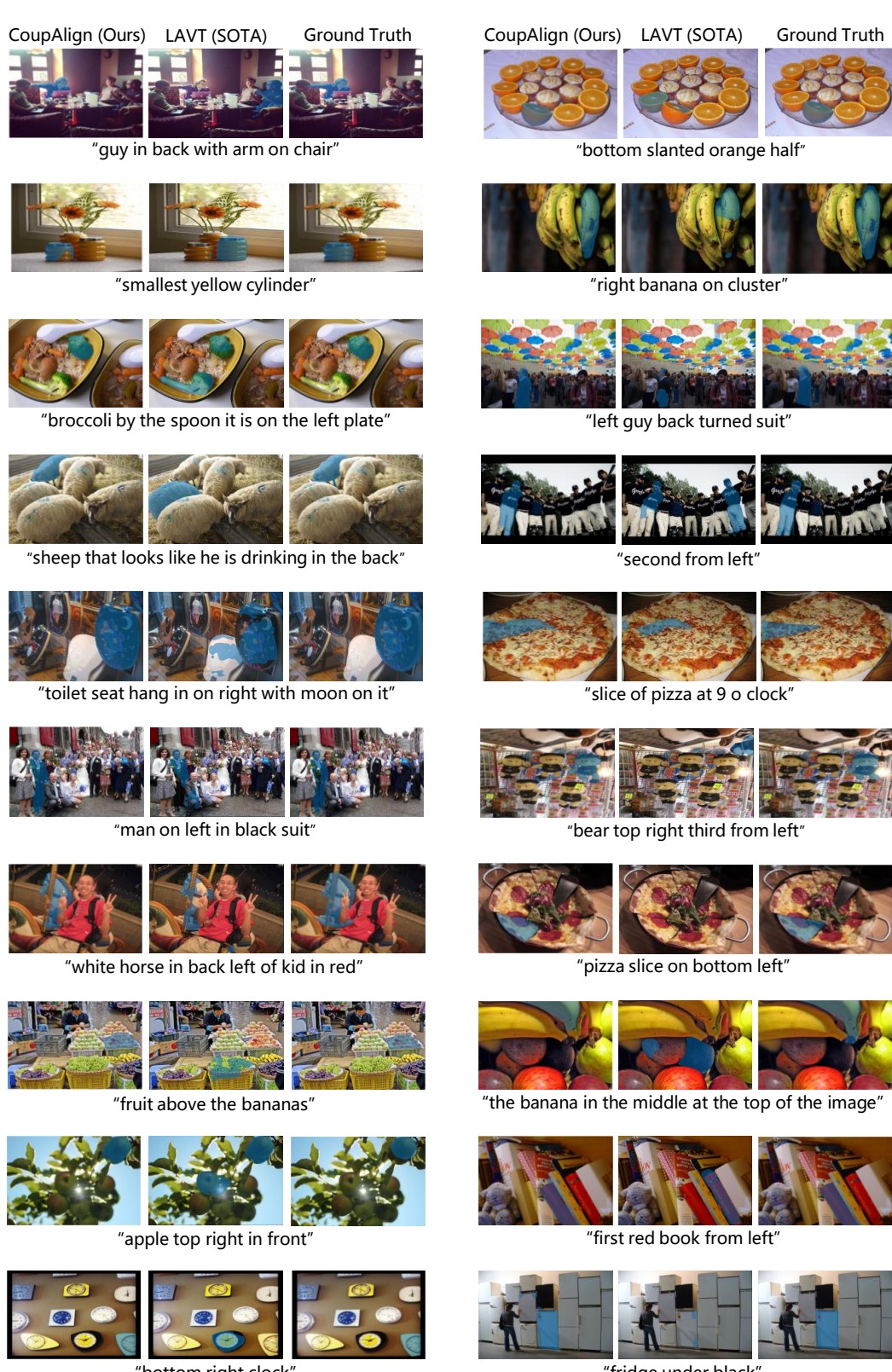

Figure 2: Visualization examples of the correct predictions of CoupAlign while LAVT typically fails.