# OpenReview forum: "CoupAlign: Coupling Word-Pixel with Sentence-Mask Alignments for Referring Image Segmentation"
_NeurIPS.cc/2022/Conference — NeurIPS 2022 Accept_

### Official Review · Reviewer_gpWt · 2022-07-01

**Rating:** 6
**Confidence:** 5
**Soundness:** 3 good
**Presentation:** 3 good
**Contribution:** 2 fair

**Summary:**

This paper proposes CoupAlign for referring image segmentation, which couples sentence-mask alignment with word-pixel alignment to enforce the mask-aware constraint. Experiments validate the effectiveness of the method.

**Questions:**

- The precision@0.5/0.7/0.9 values of the 3-rd and 4-th row are the same. Is this a typo error? Or it is just a coincidence.
- The attention weights on different words are not analyzed in the WPA module. It is better to show if the model correctly attends to the referring motion or appearance words.
- In table 4, I think there is one experiment missing, i.e., w/ Bi-WPA, w/o Uni-WPA, w/o SMA, w/o Aux Loss.
- From Figure 2 and Table 4, I think the baseline without SMA is that directly summarizes all mask query embeddings without weights generated from the sentence features. What if evaluating the importance of each mask embedding from themselves? That is, generate the weights from the mask embeddings (e.g., using MLP like the equation (3) in [3]) without the sentence embeddings.
- Could you please show the diversity of masks generate from mask queries by statistics?
- The computational cost or runtime should be discussed.
- Experiments on RefCOCO+ [4] are missing. Why?
- Why the improvement in the test split of G-Ref is so small (i.e., 0.13%) but significant on Val split (i.e., 1.6%). Does it mean the proposed methods overfit the train and Val splits?
- The improvements of WPA when inserting it into different stages of the encoders should be discussed. And what if only using the words features from the last stage of the language encoder instead of from the intermediate layers.

[3] SeqFormer: a Frustratingly Simple Model for Video Instance Segmentation, Arxiv, 2021.
[4] Modeling context in referring expressions, ECCV, 2016.


**Limitations:**

The limitations and potential negative societal impact are discussed in the supplementary files.

**Strengths And Weaknesses:**

Strengths:

- The paper is mostly clear and easy to follow.
- The writing of this paper is good and the figures are appealing.

Weaknesses

- Novelty. The basic idea of the proposed CoupAlign framework is to couple sentence-mask alignment with word-pixel alignment for consistent and accurate segmentation results. However, the proposed Word-Pixel Alignment (WPA) module is just the same as the Language-Aware Deep Fusion proposed in GLIP [1]. The Auxiliary Loss can be regarded as a simple supervised pixel-level contrastive learning loss, which has been demonstrated by recent works [2]. The proposed Sentence-Mask Alignment (SMA) module can be a contribution, but I think it is just a cross-modal version weighted-summation of queries from the DETR-decoder-like mask generator. In conclusion, the novelty of this paper is limited.

[1] Grounded Language-Image Pre-training, CVPR, 2022.

[2] Exploring Cross-Image Pixel Contrast for Semantic Segmentation, ICCV, 2021.

---

> ### Author Response · Authors · 2022-08-02
> **Response to Reviewer gpWt**
>
> Thank you for your detailed comments. We will explain the novelty and differences from other works you mentioned in details below. We sincerely hope you can recognize the significance of our work.
>
>
> ### **Novelty**
>
> Regarding the implementation of CoupAlign, the design of the core components is inspired by some existing modules in other computer vision tasks such as vision-language grounding [1], semantic segmentation [2] and object detection [3] as you mentioned in [Weaknesses]. However, directly combining these modules **CANNOT** achieve **fine-detailed word-pixel alignment** and **accurate sentence-mask alignment** (as shown in Fig.4) which are the main challenges of referring image segmentation. Here we compare our method with the works you mentioned one by one:
>
> 1) Compare with GLIP [1]: Our WPA module is different from the cross-modal feature fusion in GLIP [1] in three aspects. First, the fusion module in [1] only achieves word-region alignment, while our WPA performs word-pixel alignmemt which is more fine-grained. Second, the fusion in [1] can only align object names to visual features, while WPA can align not only object names but also attribute and position words (e.g., blue, right, second, etc.). Last, our WPA contains two Gate units (see Eq.2 and Eq.6 in our paper) to regulate the information flow between language and visual features, but in [1] the features are directly fused without any regulating and filtering.
>
> 2) Compare with loss in [2]: The main difference between our Aux Loss and the contrastive loss in [2] is that the loss in [2] enforces cross-image pixel embeddings belonging to the same semantic class to be more similar than embeddings from different classes, while our Aux Loss divides the pixels of the referred instance from others within an image no matter whether they belong to the same class or not.
>
> 3) Compare with DETR [3]: The DETR decoder enumerates all possible box proposals in an image without any guidance, while SMA generates possible mask proposals guided by the language caption. To achieve this, we further employ the language embedding as an external query for weighting the masks and summarize their locations to find the target.
>
> In summary, our modules are both well-motivated and elaborately designed for accurately segmenting object instances described by arbitrary lanaguage input.
>
> In the following we conclude **our main contributions** from four perspectives:
>
> 1) Well-designed framework components: The WPA module achieves fine-grained word-pixel alignment and can balance fusion and computation. The SMA module that achieves sentence-mask alignment is coupled together with the word-pixel alignment to form a new cross-modal alignment from local to global.
>
> 2) A novel referring segmentation framework: Existing referring segmentation methods typically ignore the semantic consistency of pixels within the same object. To address this, our CoupAlign proposes to introduce mask constraints for enhancing the cross-modal alignment and generate more accurate segmentation results. CoupAlign is general and can be easily extended to weakly supervised data like image-text pairs.
>
> 3) Strong performance: The quantitative results show that CoupAlign consistently outperforms existing SOTA methods on G-Ref, RefCOCO, RefCOCO+ and ReferIt, and the qualitative results show the remarkable ability of CoupAlign in localizing objects from crowds (see Fig.4).
>
> 4) Code release: Since there are few available code bases to reproduce existing referring segmentation methods, we will release all our source code and models to promote future research.
>
>
> [1] Grounded Language-Image Pre-training, CVPR, 2022.
>
> [2] Exploring Cross-Image Pixel Contrast for Semantic Segmentation, ICCV, 2021.
>
> [3] End-to-End Object Detection with Transformers, ECCV, 2020.

---

> > ### Author Response · Authors · 2022-08-02
> > **Response to Reviewer gpWt (Continued 1)**
> >
> > We list all your questions and respond to them point-to-point:
> >
> > * **Results in the 3-rd and 4-th row in Tab.4:** "The precision@0.5/0.7/0.9 values of the 3-rd and 4-th row are the same."
> > * **Attention weights of different words:** "The attention weights on different words are not analyzed in the WPA module. It is better to show if the model correctly attends to the referring motion or appearance words."
> > * **Remove both SMA and Aux Loss:** "In table 4, I think there is one experiment missing, i.e., w/ Bi-WPA, w/o Uni-WPA, w/o SMA, w/o Aux Loss."
> > * **SMA self-weighting baseline:** "From Figure 2 and Table 4, I think the baseline without SMA is that directly summarizes all mask query embeddings without weights generated from the sentence features."
> > * **Diversity statistics of generated masks:** "Could you please show the diversity of masks generate from mask queries by statistics?"
> > * **Computational cost:** "The computational cost or runtime should be discussed."
> > * **Experiments on RefCOCO+:** "Experiments on RefCOCO+ are missing. Why?"
> > * **Results on G-Ref:** "Does it mean the proposed methods overfit the train and Val splits?"
> > * **Ablation study of WPA module:** "The improvements of WPA when inserting it into different stages of the encoders should be discussed."
> >
> >
> > ### **Results in the 3-rd and 4-th row in Tab.4**
> >
> > We apologize for the typos of the prec@0.5/0.7/0.9 values in the 4-th row of the table. The correct values are 85.32\%, 75.31\%, and 30.14\%, respectively. We have corrected them in the revised paper.
> >
> > ### **Attention weights of different words**
> >
> > Here we show some examples of word attention weights. They correspond to the visualization examples in Fig.4 in our paper. The word-level attention weights for "child in blue shirt" are (0.5, 0.0, 0.4, 0.1), the weights for "man on right" are (0.6, 0.1, 0.3), and the weights for "lady in white dress" are (0.5, 0.0, 0.1, 0.4).
> >
> > ### **Remove both SMA and Aux Loss**
> >
> > In Tab.4 of our paper, we have conducted ablation studies to remove each one of the three components (WPA, SMA, Aux Loss) to validate their effectiveness. Here we remove both SMA and Aux Loss as you suggested, obtaining 72.84\% oIoU and 73.69\% mIoU.
> >
> > ### **SMA self-weighting baseline**
> >
> > Thanks for your suggestion. We use an MLP layer to generate weights for the mask embeddings, and summarize the mask embeddings according to the weights. The performance of this self-weighting baseline drops (74.70\% vs. 73.95\% oIoU, 75.49\% vs. 74.78\% mIoU), which demonstrates the effectiveness of the cross-modal weighting of our SMA.
> >
> >
> > ### **Diversity statistics of generated masks**
> >
> > Thanks for your suggestion. For each image, we calculate the IoU score between the generated masks and the ground-truth mask, and then count the number of masks according to the IoU scores to see the mask diversity. In the following table, we summarize the mask numbers at each IoU ranges on the test set of RefCOCO. We can observe that the distribution of the numbers of masks is approximately uniform over different IoU levels, which demonstrates the diversity of the generated masks.
> >
> > | IoU range         | 0-0.2            | 0.2-0.5 | 0.5-0.7    |  0.7-1.0    |
> > | --------------- | --------------- | ----- | --------------- |------|
> > | percentage (\%)        | 25           | 10 |  27           |      38  |
> >
> > ### **Computational cost**
> >
> > We test the inference time of our CoupAlign and the most recent SOTA method LAVT on a NVIDIA V100 GPU. CoupAlign costs 38ms per image on average and LAVT costs 41ms, which demonstrates the computational efficiency of CoupAlign.
> >
> > ### **Experiments on RefCOCO+**
> >
> > As shown in the following table, the performance of CoupAlign on RefCOCO+ is comparable to or better than LAVT which is the most recent state-of-the-art method (published in CVPR2022). The improvements on RefCOCO+ are less significant than those on G-Ref and RefCOCO. This is because the language captions in RefCOCO+ hardly contain the descriptions of the relative or absolute spatial locations (e.g., "closest" or "right bottom") of the target object in images, and the effectiveness of our method mainly lies in the ability of localizing objects in such challenging scenarios, such as "right first bottom fridge" and "middle row second kid from right" in Fig.3 of our paper. The new experimental results have been added to the supplementary materials.
> >
> > | Method          | val             | testA | testB           |
> > | --------------- | --------------- | ----- | --------------- |
> > | ReSTR[4]        | 55.78           | 60.44 | 48.27           |
> > | CRIS[6]         | 62.27           | 68.08 | 53.68           |
> > | LAVT[7]         | 62.14           | **68.38** | 55.10           |
> > | CoupAlign(Ours) | **62.92** | 68.34 | **56.69** |
> >
> >
> > ### **Results on G-Ref**
> >
> > In our experiment, the role of validation set is the same as that of the test set. The performance difference may come from their different data distributions.

---

> > > ### Author Response · Authors · 2022-08-03
> > > **Response to Reviewer gpWt (Continued 2)**
> > >
> > >
> > > ### **Ablation study of WPA module**
> > >
> > > Great suggestion! First of all, we insert a WPA module at different encoding stages. As shown in the table below, the WPA module at the 4-th stage is more effective than those inserted at other stages. In our experiment, we use four WPA modules, two of which are in the early encoding stage and the other two are in the late encoding stage. We also conduct two baseline models that alternatively remove two WPA modules at early or late encoding stages. As shown in the following table, when we remove the last two WPA modules the performance drops about 2\% (74.70\% vs. 72.74\% oIoU, 75.49\% vs. 73.87\% mIoU), and when we remove the first two WPA modules the performance drops about 0.8\% (74.70\% vs. 73.61\% oIoU, 75.49\% vs. 74.68\% mIoU). These results validate the effectiveness of WPA modules at both early and late stages and indicate that the latter WPA modules play a more important role in our model.
> > >
> > > | WPA's number | WPA's position | oIoU  | mIoU  |
> > > | ------------ | -------------- | ----- | ----- |
> > > | 4            | stage 1,2,3,4  | **74.70** | **75.49** |
> > > | 2            | stage 1,2      | 72.74 | 73.87 |
> > > | 2            | stage 3,4      | 73.93 | 74.88 |
> > > | 1            | stage 4        | 73.61 | 74.68 |
> > > | 1            | stage 3        | 72.53 | 73.47 |
> > > | 1            | stage 2        | 72.63 | 73.48 |
> > > | 1            | stage 1        | 72.59 | 73.97 |

---

> > > > ### Author Response · Authors · 2022-08-08
> > > > **Response to Reviewer gpWt**
> > > >
> > > > Dear reviewer,
> > > >
> > > > We have tried to address your concerns and all your questions in our earlier responses. If you have any additional questions or suggestions, we are very happy to discuss with you.

---

> > > > > ### Comment · Reviewer_gpWt · 2022-08-10
> > > > > **Thanks for your response**
> > > > >
> > > > > Thanks for solving my concerns with new experiments and detailed explanation. And I am glad to raise my rating up for accepting the paper.

---

> > > > > > ### Author Response · Authors · 2022-08-10
> > > > > > **Thanks**
> > > > > >
> > > > > > Dear reviewer,
> > > > > >
> > > > > > Thank you very much for your support!

---

### Official Review · Reviewer_Vuxv · 2022-07-10

**Rating:** 5
**Confidence:** 4
**Soundness:** 3 good
**Presentation:** 3 good
**Contribution:** 2 fair

**Summary:**

This paper proposes a new model for referring image segmentation (RIS). The model takes in an image and a natural language description as input, and outputs the segmentation mask corresponding to the language input. The proposed approach features 1) a hierarchical fusion architecture in its encoder to gradually fuse information from the visual and language inputs (dubbed as Word-Pixel Alignment or WPA), and 2) a module to assemble segmentation masks from a set of candidate masks and the language representation (dubbed as Sentence-Mask Alignment or SMA). In terms of experiments, the authors train and test their models for RIS on two datasets -- RefCOCO and G-Ref, and achieved competitive results compared to previous SOTA methods. The authors also carried out ablation studies on various design choices and provided some qualitative visualizations to better understand the model's behavior.


**Questions:**

- In Eq 5, V'_i is "assembled" from the language feature while L'_i is assembled from the visual feature? Should these two be swapped?
- In Eq 14, is y^-_k sampled from the same image or from some other images?
- L287, "when SMA is removed", how do you remove SMA while still generating the segmentation conditioned on the language input? This requires more elaboration.


**Limitations:**

Yes

**Strengths And Weaknesses:**

Strengths
+ The task of segmenting images with arbitrary language input is interesting and could be a prominent task in near future.
+ For presentation, 1) the illustration in Fig. 2 is comprehensive and helpful and 2) the visualization in Fig. 4 is very helpful for readers to better understand the model behavior.
+ Overall, the proposed model is intuitive and achieves strong performance when compared to other SOTA methods on popular datasets.

Weaknesses
- Notations are confusing. For example, there are d_k, d_s, d_q, are they equal to each other? In Eq 11, L_g is with shape 1 x D while Q_o is with N x d_q, does this imply D == d_q? Also, in Eq 12, Y_1 is with shape H/4 x W/4 x d_s while Q_o is with size N x d_q, how can you multiply these two (unless d_q == d_s, which I did not see stated anywhere)?
- Since both WPA and Cross Attention are fusing the visual and language information, why using different attention computation for these two? The design seems arbitrary and I did not see any justification for the difference.
- Fig 3, are these examples representative or through cherry picking? Since I did not see why the proposed method would have a qualitatively different behavior compared to previous models (e.g. LAVT) by adding more intermediate level fusion. If this is not representative, I think it's better to not include these as they can be misleading (i.e. to make readers think the proposed model is qualitatively better/different).
- L281-6, since this is where the main technical novelty lie, it would be better to have more detailed ablations on questions like how many layers of fusion is needed? Or which fusion layer contributes the most?

############# POST REBUTTAL NOTE #############
The authors have properly addressed my comments above with the added experiments and paper revisions. With this, I will raise up my rating to accept the paper.

---

> ### Author Response · Authors · 2022-08-02
> **Response to Reviewer Vuxv**
>
> Thanks for your detailed and constructive comments. We are glad to see that you appreciated our research on the interesting problem of segmenting images with arbitrary language input, our illustration and visualization, and the strong performance.
> We believe that your main concern and rating are from some confusing notations in our manuscript. In the following we will address these issues one-by-one and we have clarified your concerns in the revised paper. We sincerely hope you can recognize the significance of our work. First of all, we list your questions:
>
> * **Notation clarification:** "Notations are confusing."
> * **Difference between WPA and CrossAttn:** "Since both WPA and Cross Attention are fusing the visual and language information, why using different attention computation for these two?"
> * **About Fig.3:** "Fig 3, are these examples representative or through cherry picking?"
>
> * **Ablation study of WPA module:** "it would be better to have more detailed ablations on questions like how many layers of fusion is needed? Or which fusion layer contributes the most?"
> * **Questions about equations:** "Eq.5", "Eq.14", and "L287".
>
> ### **Notation clarification**
>
> We apologize for the confusing notations.
> 1) Sorry for the typo. $d_k$ is equal to $d$. In our experiment, $d_s$ is set to 256 and $d_q$ is set to 512. They are not equal.
> 2) In Eq.11, $Q_o$ is first projected via a fully-connected layer from $N \times d_q$ to $N \times D$, and then multiplied with $L_g$.
> 3) In Eq.12, similar to $Q_o$, $Y_1$ is projected from $\frac{H}{4} \times \frac{W}{4} \times d_s$ to $\frac{H}{4} \times \frac{W}{4} \times D$ and then multiplied with $Q_o$.
>
> ### **Difference between WPA and CrossAttn**
>
> The WPA module is implemented based on bidirectional cross-attention to integrate both language information into visual embeddings and visual information into language embeddings. In contrast, CrossAttn in Eq.8 is a kind of unidirectional attention which only integrates language information into visual embeddings.
>
> ### **About Fig.3**
>
> We visualize the representative examples where the target objects are correctly localized by CoupAlign but typically missed by LAVT. The localization ability of CoupAlign is superior over LAVT, because 1) the word-pixel alignment exchanges information between visual and language encoders to generate rich and compact pixel-level embeddings and better understands the language captions, 2) the sentence-mask alignment built upon the well aligned pixel-level embeddings introduces mask constraints for the cross-modal alignment. In Tab.2, we can see that in different IoU ranges, the number of failure cases of CoupAlign is less than that of LAVT, which indicates that the masks predicted by CoupAlign have greater overlap with the ground-truth, and are more accurate than those of LAVT.
>
>
> ### **Ablation study of WPA module**
>
> Thanks for your valuable suggestion. In our experiment, we use four WPA modules, two of which are in the early encoding stage and the other two are in the late encoding stage. To study the effects of the numbers of WPA modules, we first conduct two baseline models that alternatively remove two WPA modules at early or late encoding stages. As shown in the following table, when we remove the last two WPA modules the performance drops about 2\% (74.70\% vs. 72.74\% oIoU, 75.49\% vs. 73.87\% mIoU), and when we remove the first two WPA modules the performance drops about 0.8\% (74.70\% vs. 73.61\% oIoU, 75.49\% vs. 74.68\% mIoU). These results validate the effectiveness of WPA modules at both early and late stages and indicate that the latter WPA modules play a more important role in our model. Then, we only use one WPA module, which is inserted at different encoding stages. As shown in the table below, the WPA module at the 4-th stage is more effective than those inserted at other stages.
>
> | WPA's number | WPA's position | oIoU  | mIoU  |
> | ------------ | -------------- | ----- | ----- |
> | 4            | stage 1,2,3,4  | **74.70** | **75.49** |
> | 2            | stage 1,2      | 72.74 | 73.87 |
> | 2            | stage 3,4      | 73.93 | 74.88 |
> | 1            | stage 4        | 73.61 | 74.68 |
> | 1            | stage 3        | 72.53 | 73.47 |
> | 1            | stage 2        | 72.63 | 73.48 |
> | 1            | stage 1        | 72.59 | 73.97 |
>
> ### **Questions about equations**
>
> 1) Eq.5: $V'_i$ and $L'_i$ in Eq.5 should be swapped.
>
> 2) Eq.14: The negative samples $y_k^-$ are sampled from the same image.
>
> 3) L287: Because the calculation of SMA depends on the mask generator's output, when SMA is removed, the mask generator is also removed, and an MLP layer is directly used to the output of SegHead $Y_1$ to obtain the final prediction.

---

> > ### Comment · Reviewer_Vuxv · 2022-08-07
> > **Thanks for the response**
> >
> > I thank the authors for their responses.
> >
> > - Difference between WPA and CrossAttn
> > The response regarding this point is still not fully satisfactory, since I was asking about the justification of using two different types of attentions modules. It would be nice to have an experiment ablating on this by applying either WPA and CrossAttn throughout.
> >
> > - About Fig.3
> > I understand conceptually why the proposed method might produce more accurate delineation of object boundaries. However, as I expressed in my original question, I wasn't sure this translates to qualitative difference like in the case of "man far right second step" or "right first bottom fridge", or "whole bear next to empty spot on center seat". Therefore, I want to know whether this is a widespread qualitative difference that the authors observe, or these are more of a cherry picked results (which is fine, but you might not want to include too many examples like this, as that leads people to think it's a general phenomenon)
> >
> > - Ablation study of WPA module
> > Thanks for the explanation, this is helpful for better understanding the approach.

---

> > > ### Author Response · Authors · 2022-08-08
> > > **Response to Reviewer 3 (Vuxv)**
> > >
> > > Dear reviewer, thank you very much for your reply!
> > >
> > > ### **Difference between WPA and CrossAttn**
> > >
> > > As you suggested, we further conduct two experiments and show the results in the table below.
> > >
> > > 1) When we replace CrossAttn with WPA the performance slightly drops (74.70% vs. 74.17% oIoU, 75.49% vs. 75.02% mIoU). The reason is that the CrossAttn is multi-head attention while WPA is single-head. In general, multi-head attention is more effective than single-head attention since it can jointly attend multiple positions.
> > >
> > > 2) When we replace WPA with CrossAttn in our model, the performance is sligntly lower than CoupAlign (74.70% vs. 74.32% oIoU, 75.49% vs. 74.92% mIoU). The single-head WPA modules are changed into CrossAttn (which is multi-head) but does not cause performance improvement, because CoupAlign encoder uses multiple WPA modules such that the multi-layer single-head attention can also attend multiple positions. Moreover, increasing the attention heads of WPA can also increase the model complexity, which may harm the model robustness and cause slight damage to the performance.
> > >
> > > | Experimental Settings | oIoU  | mIoU  |
> > > | ------------------- | ----- | ----- |
> > > | CoupAlign (Ours)     | **74.70** | **75.49** |
> > > | Replace CrossAttn with WPA   | 74.17 | 75.02 |
> > > | Replace WPA with CrossAttn   | 74.32 | 74.92 |
> > >
> > >
> > > ### **About Fig.3**
> > >
> > > We agree with you that showing too many such examples may mislead the readers.
> > > In Fig.3 of the revised paper, we have deleted some similar examples, and no longer compare specific cases with SOTA to show the superiority of our method. The new figure aims to illustrate the remarkable capability of our method in accurately segmenting the target object from crowds. What's more, we show failure cases of CoupAlign to analyze the limitation.
> > >
> > > ### **Ablation study of WPA module**
> > >
> > > Thanks for your recognization! We have added the experimental results in the revised paper.

---

> > > > ### Comment · Reviewer_Vuxv · 2022-08-09
> > > > **Thanks for the response**
> > > >
> > > > Thanks for the new experiments on comparing WPA and CrossAttn, it's assuring to see that the specific instantiation of the attention module does not matter much.
> > > >
> > > > Overall, I appreciate the author's efforts and I'm satisfied with the author's response to my questions and concerns. I'm happy to raise my rating up to accept the paper.

---

> > > > > ### Author Response · Authors · 2022-08-10
> > > > > **Thanks**
> > > > >
> > > > > Dear reviewer,
> > > > >
> > > > > Thank you very much for your support!

---

### Official Review · Reviewer_hX8h · 2022-07-12

**Rating:** 5
**Confidence:** 4
**Soundness:** 3 good
**Presentation:** 3 good
**Contribution:** 3 good

**Summary:**

In order to employ the semantic consistency of pixels within the same object, this paper proposes a CoupAlign mechanism to couple a work-pixel alignment (WPA) and a sentence-mask alignment (SMA). WPA fuses the linguistic and pixel-level features within the intermediate layers of the feature encoders. SMA weights the generated masks to localize the referred object. In addition, the authors provide an auxiliary contrastive loss for facilitating the segmentation accuracy. The experiments show that the proposed CoupAlign mechanism achieves state-of-the-art referring image segmentation performance.

**Questions:**

Though the proposed CoupAlign mechanism coupling WPA and SMA is interesting, the manuscript lacks some clear terminology definitions, as mentioned in [Weaknesses], and the experiments are not supported by sufficient standard evaluating datasets. I would like to see more information in the authors’ response.

**Limitations:**

 The authors adequately addressed the limitations and potential negative societal impact of their work.

**Strengths And Weaknesses:**

[Strengths]
+ The idea of implementing the CoupAlign mechanism is interesting.
+ The manuscript is well organized, and most paragraphs are easy to follow.
+ The references are adequate.

[Weaknesses]
- The experiment misses evaluating standard datasets, ReferIt and RefCOCO+, for comparison with other RIS methods. Such an incomplete comparison makes the evaluating experiments weak.
- Missing some function descriptions harms the readers to reimplement the proposed model. For example, what are the functions of SwinStage and BERTStage in (3)? It is unclear why these functions relate to two different stages. Also, what is the function of CrossAttn in (8) and MaskGenerator in (9)? Please explain these functions' structures mentioned above, including the inputs and outputs, to clarify the model design.
- The shapes of the projection matrices mentioned in section 3.2 are not correct.

---

> ### Author Response · Authors · 2022-08-02
> **Response to Reviewer hX8h**
>
> We appreciate that you recognize the significance of our work. We will respond to your concerns in the following:
>
> * **Evaluation on more datasets:** "The experiment misses evaluating standard datasets, ReferIt and RefCOCO+, for comparison with other RIS methods."
>
> * **Function descriptions:** "Missing some function descriptions harms the readers to reimplement the proposed model."
>
> * **Projection matrices in section 3.2:** "The shapes of the projection matrices mentioned in section 3.2 are not correct."
>
> ### **Evaluation on more datasets**
>
> **1. ReferIt:** As shown in the table below, the performance of CoupAlign on ReferIt is higher than the state-of-the-art method ReSTR, which is published in CVPR2022.
>
> | Method    | test            |
> | --------- | --------------- |
> | LCSM [1]   | 66.57           |
> | EFN [2]    | 66.70           |
> | ReSTR [3]  | 70.18           |
> | CoupAlign (Ours) | **73.28**       |
>
>
>
> **2. RefCOCO+:** As shown in the following table, the performance of CoupAlign on RefCOCO+ is comparable to or better than LAVT which is the most recent state-of-the-art method (published in CVPR2022). The improvements on RefCOCO+ are less significant than those on G-Ref and RefCOCO. This is because the language captions in RefCOCO+ hardly contain the descriptions of the relative or absolute spatial locations (e.g., "closest" or "right bottom") of the target object in images, and the effectiveness of our method mainly lies in the ability of localizing objects in such challenging scenarios, such as "right first bottom fridge" and "middle row second kid from right" in Fig.3 of our paper.
>
>
> | Method          | val             | testA | testB           |
> | --------------- | --------------- | ----- | --------------- |
> | ReSTR [3]        | 55.78           | 60.44 | 48.27           |
> | CRIS [4]         | 62.27           | 68.08 | 53.68           |
> | LAVT [5]         | 62.14           | **68.38** | 55.10       |
> | CoupAlign (Ours) | **62.92**       | 68.34 | **56.69**       |
>
> These new experimental results have been added to the supplementary materials.
>
> ### **Function descriptions**
>
> Sorry for the confusion. The SwinStage represents the network stages in Swin Transformer [6] and consists of linear embedding or patch merging layers together with Swin Transformer Blocks. Our visual encoder has four SwinStage. As is described in BERT [7], each of the three hidden layers are referred to as a BERTStage. Our language encoder is a BERT-BASE model which has 12 hidden layers and the layers can be divided into four BERTStage. WPA modules can be inserted after each of the SWinStage and BERTStage. CrossAttn represents a plain multi-head self-attention layer that takes the concatenated vision and language features $\mathbb{R}^{(H_oW_o+T)\times D}$ as inputs and outputs the vision features $\mathbb{R}^{H_oW_o\times D}$. MaskGenerator is a 6-layer transformer decoder, $S_o$ is the key and the value, and $Q$ is the query.
>
> ### **Projection matrices in section 3.2**
>
> We apologize for the typo in Line 149 of the original manuscript, where the shape of the $W_i^l$ should be $D\times d$, but not $D\times d_k$.
>
> ___
> **Reference**
>
> [1] Linguistic structure guided context modeling for referring image segmentation. ECCV, 2020.
>
> [2] Encoder fusion network with co-attention embedding for referring image segmentation. CVPR, 2021.
>
> [3] Restr: Convolution-free referring image segmentation using transformers. CVPR, 2022.
>
> [4] Cris: Clip-driven referring image segmentation. CVPR, 2022.
>
> [5] Lavt: Language-aware vision transformer for referring image segmentation. CVPR, 2022.
>
> [6] Bert: Pre-training of deep bidirectional transformers for language understanding. ACL, 2018.

---

> > ### Comment · Reviewer_hX8h · 2022-08-08
> > **Thanks for the authors' response.**
> >
> > ### Thanks for the authors' response.
> > * It is better to integrate the additional experiments of ReferIt and RefCOCO+ in Table 1 (main paper).
> > * Function descriptions are more precise than the previous version.
> > * It is better to clarify the difference between WPA and CrossAttn, as recommended by the reviewer Vuxv.

---

> > > ### Author Response · Authors · 2022-08-08
> > > **Response to Reviewer 2 (hX8h)**
> > >
> > > Dear reviewer, thank you very much for your reply!
> > >
> > > ### **ReferIt and RefCOCO+**
> > >
> > > Good suggestion! We have added the experimental results of ReferIt and RefCOCO+ in the revised paper.
> > >
> > > ### **Difference between WPA and CrossAttn**
> > >
> > > We further conduct two experiments and show the results in the table below.
> > >
> > > 1) When we replace CrossAttn with WPA the performance slightly drops (74.70% vs. 74.17% oIoU, 75.49% vs. 75.02% mIoU). The reason is that the CrossAttn is multi-head attention while WPA is single-head. In general, multi-head attention is more effective than single-head attention since it can jointly attend multiple positions.
> > >
> > > 2) When we replace WPA with CrossAttn in our model, the performance is sligntly lower than CoupAlign (74.70% vs. 74.32% oIoU, 75.49% vs. 74.92% mIoU). The single-head WPA modules are changed into CrossAttn (which is multi-head) but does not cause performance improvement, because CoupAlign encoder uses multiple WPA modules such that the multi-layer single-head attention can also attend multiple positions. Moreover, increasing the attention heads of WPA can also increase the model complexity, which may harm the model robustness and cause slight damage to the performance.
> > >
> > > | Experimental Settings | oIoU  | mIoU  |
> > > | ------------------- | ----- | ----- |
> > > | CoupAlign (Ours)     | **74.70** | **75.49** |
> > > | Replace CrossAttn with WPA   | 74.17 | 75.02 |
> > > | Replace WPA with CrossAttn   | 74.32 | 74.92 |

---

### Official Review · Reviewer_gikD · 2022-07-14

**Rating:** 6
**Confidence:** 3
**Soundness:** 3 good
**Presentation:** 3 good
**Contribution:** 2 fair

**Summary:**

This paper proposed a cross-modal alignment model (CoupAlign) for referring image segmentation.  The authors propose to use coupled sentence-mask alignments with word-pixel alignment to enforce the model learned more accurate and consistent segmentation masks. Word-pixel alignment serves as an early fusion module that fuses the features between each block of swing transformer feature and BERT feature. The Sentence mask alignment learns to weight the mask using the sentence embedding to localize the referred object. The authors benchmark the proposed method on 2 refer segmentation benchmarks and achieve state-of-the-art performance.

**Questions:**

- Is it possible to extend the current framework to incorporate weakly supervised data such as image caption pairs?

- What is the role of WPA module and how many do we need it. What is the good balance between fusion and computing?

- For cross attention, why not use multi-head attention? is there any specific reason?


**Limitations:**

Yes

**Strengths And Weaknesses:**

[Strength]

- WPA module serves as an early fusion module that can balance the fusion and compute.

-  The sentence mask alignment (SMA) module is a general module that can scale up to weakly supervised learning such as GLID.

- The proposed model achieves state-of-the-art performance on refer segmentation benchmark.

- The visualization of word-pixel alignment shows the effectiveness of the module and aux loss.

[Weakness]

- It would be good to conduct an ablation study on how many WPA module is needed in the model. Does the early fusion (first few blocks of the model) need the WPA module or the more WAP module the better?

- It seems the WPA module use cross-attention similar to [1], so it would be good to compare the difference between different cross-attention operations. Why not use multi-head attention on the WAP module?

[1] Lu, J., Yang, J., Batra, D. and Parikh, D., 2016. Hierarchical question-image co-attention for visual question answering. Advances in neural information processing systems, 29.

---

> ### Author Response · Authors · 2022-08-02
> **Response to Reviewer gikD**
>
> We are grateful for your comprehensive and encouraging review. We are pleased that you appreciate the technical contributions, the state-of-the-art performance, and the effectiveness of our modules and aux loss. In the following, we will respond to your concerns and questions:
>
>
> * **Role of WPA and ablation of WPA module:** "What is the role of WPA module and how many do we need it.", "What is the good balance between fusion and computing?"
> * **Compare with different cross-attention operations:** "so it would be good to compare the difference between different cross-attention operations. Why not use multi-head attention on the WAP module?"
> * **Extend to weakly supervised data:** "Is it possible to extend the current framework to incorporate weakly supervised data such as image caption pairs?"
>
> ### **Role of WPA and ablation of WPA module**
>
> 1) Role of WPA: Our model uses four WPA modules to achieve cross-modal alignment from local to global. Such elaborate alignments generate rich and compact embeddings at both the modalities.
>
> 2) Ablation study of WPA module: In our experiment, we use four WPA modules, two of which are in the early encoding stage and the other two are in the late encoding stage. To study the effects of the numbers of WPA modules, we first conduct two baseline models that alternatively remove two WPA modules at early or late encoding stages. As shown in the following table, when we remove the last two WPA modules the performance drops about 2\% (74.70\% vs. 72.74\% oIoU), and when we remove the first two WPA modules the performance drops about 1\% (74.70\% vs. 73.61\% oIoU). These results validate the effectiveness of WPA modules at both early and late stages and indicate that the latter WPA modules play a more important role in our model. Then, we only use one WPA module, which is inserted at different encoding stages. As shown in the table below, the WPA module at the 4-th stage is more effective than those inserted at other stages.
>
> 3) Balance between fusion and computing: When we remove all the WPA modules, the inference time reduces from 38ms to 34ms per image on a single NVIDIA V100 GPU, and the performance drops a lot, i.e., from 74.70\% to 70.43\% oIoU (see Tab.4 in our paper). Therefore, WPA module can bring significant performance gains with a small computational cost.
>
>
> | WPA's number | WPA's position | oIoU  | mIoU  |
> | ------------ | -------------- | ----- | ----- |
> | 4            | stage 1,2,3,4  | **74.70** | **75.49** |
> | 2            | stage 1,2      | 72.74 | 73.87 |
> | 2            | stage 3,4      | 73.93 | 74.88 |
> | 1            | stage 4        | 73.61 | 74.68 |
> | 1            | stage 3        | 72.53 | 73.47 |
> | 1            | stage 2        | 72.63 | 73.48 |
> | 1            | stage 1        | 72.59 | 73.97 |
>
> ### **Compare with different cross-attention operations**
>
>
> Our WPA module is implemented based on the bidirectional cross-attention (BiAttn) module. The parallel co-attention module in [1] is similar to BiAttn but uses more learnable parameters. To compare with other attention variants, we replace BiAttn with the unidirectional attention (UniAttn) module and obtain lower results (74.70\% vs. 72.70\% oIoU, 75.49\% vs. 73.44\% mIoU, see Tab.4 in our paper). The reason is that UniAttn can only combine language information into the visual encoder but cannot exchange information between the two modalities.
>
> Moreover, if we change BiAttn into the multi-head version (four attention heads), the performance is slightly lower than the single-head counterpart (74.70\% vs. 74.32\% oIoU, 75.49\% vs. 74.92\% mIoU). The reasons include: 1) the effectiveness of multi-head attention stems from the ability of jointly attending to multiple positions, and the multi-layer single-head attention modules in our model can also attend to multiple positions; 2) increasing the attention heads of WPA can also increase the model complexity, which may harm the model robustness and cause slight damage to the performance.
>
> | WPA attention types | oIoU  | mIoU  |
> | ------------------- | ----- | ----- |
> | Bi-Attn             | **74.70** | **75.49** |
> | Uni-Attn            | 72.70 | 73.44 |
> | Bi-MultiHeadAttn    | 74.32 | 74.92 |
>
> [1] Hierarchical question-image co-attention for visual question answering. NeurIPS, 2016.
>
>
> ### **Extend to weakly supervised data**
>
> Thanks for your valuable suggestion. CoupAlign can be trained in two ways with weak supervision data. 1) Do not change the existing framework: directly use the pseudo masks generated by conventional weakly supervised segmentation approaches as ground-truth masks. 2) Only small changes to the loss functions: before calculating the binary cross entropy loss, a global pool layer is added, and the loss is changed into the image classification loss. Then, change the auxiliary loss to a self-supervised version, i.e., first cluster the samples into groups and then divide the positive and negative samples.

---

### Author Response · Authors · 2022-08-02
**Summary of response -- thanks to all reviewers for their thorough and constructive comments**

We thank all the reviewers for their time, insightful suggestions, and valuable comments.

We respond to each reviewer's comments in detail below. We have also revised the manuscript according to reviewer's suggestions, and we believe this makes our paper much stronger. The main changes we made include:

* In Section 3 of the revised paper, we correct typos and add notation clarifications.
* In Table 4 of the revised paper, we correct the values in the 4-th row.
* In Table 1 of the revised paper, we add the experimental results on two more datasets, i.e., ReferIt and RefCOCO+.
* In Table 5 of the revised paper, we add the ablation study on the number and position of WPA modules.

In the revised manuscripts, we have marked the revisions in blue.

---

### Meta-Review · Area_Chair_y9L4 · 2022-08-23

**Recommendation:** Accept
**Confidence:** Certain

**Metareview:**

The paper was reviewed by four reviewers and received all positive scores at the end: 2 x Borderline Accepts and 2 x Weak Accepts. Most initial concerns with the paper were with exposition and experimental validation. These concerns, however, were addressed convincingly during the rebuttal period with additional experiments and ablations, as well as direct edits to the manuscript itself. In the current form the paper would be a valuable contribution to NeurIPS program.

**Award:**

No

---

### Decision · Program_Chairs · 2022-09-14

Accept